# Eliminating VAE for Fast and High-Resolution Generative Detail Restoration

**Yan Wang, Shijie Zhao**\*, **Junlin Li, Li Zhang**
ByteDance
`{wangyan.my, zhaoshijie.0526}@bytedance.com`

## Abstract

Diffusion models have attained remarkable breakthroughs in the real-world super-resolution (SR) task, albeit at slow inference and high demand on devices. To accelerate inference, recent works like GenDR adopt step distillation to minimize the step number to one. However, the memory boundary still restricts the maximum processing size, necessitating tile-by-tile restoration of high-resolution images. Through profiling the pipeline, we pinpoint that the variational auto-encoder (VAE) is the bottleneck of latency and memory. To completely solve the problem, we leverage pixel-(un)shuffle operations to eliminate the VAE, reversing the latent-based GenDR to pixel-space GenDR-Pix. However, upscale with $\times 8$ pixelshuffle may induce artifacts of repeated patterns. To alleviate the distortion, we propose a multi-stage adversarial distillation to progressively remove the encoder and decoder. Specifically, we utilize generative features from the previous stage models to guide adversarial discrimination. Moreover, we propose random padding to augment generative features and avoid discriminator collapse. We also introduce a masked Fourier space loss to penalize the outliers of amplitude. To improve inference performance, we empirically integrate a padding-based self-ensemble with classifier-free guidance to improve inference scaling. Experimental results show that GenDR-Pix performs $2.8\times$ acceleration and 60% memory-saving compared to GenDR with negligible visual degradation, surpassing other one-step diffusion SR. Against all odds, GenDR-Pix can restore **4K** image in only **1 second** and **6 GB**.

## 1 Introduction

Super-resolution (SR) is widely exploited to reconstruct missing high-quality (HQ) details from the degraded low-quality (LQ) image for better visual experiences. Existing studies (Dong et al., 2016; Lim et al., 2017; Wang et al., 2022a; Liang et al., 2021; Wang et al., 2024c) have gained a comprehensive understanding of synthesized degradation, *e.g.*, bicubic downscaling. To develop generality and visual quality under real-world scenarios, prevalent researches (Ledig et al., 2017; Zhang et al., 2021; Wang et al., 2024b) exploit generative models to recover structural information and replenish details. Recently, diffusion models have become the new paradigm for text-to-image (Rombach et al., 2022b; Esser et al., 2024) and image-to-image tasks (Brooks et al., 2023; Wang et al., 2024b), bringing a new trend for the SR task. Nevertheless, diffusion-based SR methods struggle with slow inference and intensive memory consumption, restricting their usage in real-world scenes.

To accelerate inference of diffusion SR, recent work (Xie et al., 2024; Wu et al., 2024b; Zhang et al., 2024; Yue et al., 2024a) has discovered one-step diffusion via step distillation to minimize the overhead of diffusion inversion processing. Despite achieving the theoretically lowest complexity, existing one-step diffusion SRs easily exceed the memory upper bound for high-resolution input, requiring to crop into tiles, which is executed in serial and results in an unavoidable increase in elapsed time. Thus, reducing both runtime and memory footprint is of equal importance for practical employment. More recent attempts (Hu et al., 2024; Chen et al., 2025a) explore structural pruning or distillation to simplify or remove complex modules. They have put a lot of effort into simplifying the variational auto-encoder (VAE) since it costs more memory and even longer time in a few-step diffusion inference. However, these trials still execute the reconstruction from latent to pixel,

---

\*Corresponding Author.

simplifying VAE and intensifying the dilemma between generation and reconstruction, particularly in the SR task. For instance, eliminating the attention of the decoder carries a harmful impact on high-frequency details, leading to texture loss and sharpness decrease. Hence, the core problem of further slimming diffusion-based SR is to *replace the VAE with simpler fidelity-guaranteed operations*.

We revisit LDM (Rombach et al., 2022a) employing latent space rather than pixel space (Ho et al., 2020), *i.e.*, to reduce training and inference consumption by representing images with smaller features. However, there is a similar operation in the pixel space, *i.e.* pixel-shuffle. Existing SR models (Shi et al., 2016; Wang et al., 2022b) use pixel-shuffle operation to upsample in the last stage, which allows most computations to be performed under the smaller shape. Inspired by Chen et al. (2025a;b) and the similar role between VAE and pixel(un)shuffle, we unveil a new path to lighten the diffusion SR model, which reverses the latent-based diffusion SR to pixel-based by replacing VAE with pixel-unshuffle and pixel-shuffle operation.

Despite its intuitiveness, eliminating VAE faces several problems. Firstly, VAE conducts large-scale representation compression for input images. Nevertheless, the model with the same factor pixelshuffle hardly converges and will cause visual artifacts, such as checkerboard and repeated pattern artifacts. The pixel-space model can barely learn from latent-based generative priors without VAE. To overcome these drawbacks, this work introduces both training and inference innovations.

- To eliminate the VAE without heavy performance drops, we introduce multi-stage adversarial distillation that gradually replaces the encoder and decoder with pixel-unshuffle and pixel-shuffle operations. In the first stage, we remove the encoder by utilizing latent matching and adversarial learning on features extracted by GenDR. In the second stage, we remove the decoder by using the stage-I model as the discriminator. Moreover, we propose random padding (RandPad) augmentation to avoid discriminator collapse and improve diversity by randomly padding SR and HQ images.
- To alleviate artifacts arising from large-scale pixel(un)shuffle, we propose masked Fourier space loss to impose a penalty for anomalous spike amplitudes.
- To improve inference performance, we propose padding-based classifier-free guidance (PadCFG), which empirically integrates the self-ensemble strategy and classifier-free guidance.

Through applying these strategies to remove the VAE of the recent state-of-the-art GenDR (Wang et al., 2025), we obtain an end-to-end model, named GenDR-Pix, which directly restores high-quality images in shuffled pixel space. Compared to the GenDR prototype, GenDR-Pix achieves about $2.8\times$ acceleration and saves over 60% GPU memory.

## 2 RELATED WORK

### 2.1 GENERATIVE IMAGE SUPER-RESOLUTION

Since pioneer works (Lim et al., 2017; Zhang et al., 2018b; Wang et al., 2024c) have exhaustively explored the potential of deep-learning-based methods in the classic SR task with synthesised LQ, prevailing research (Ledig et al., 2017; Cai et al., 2019; Zhang et al., 2021) concentrates on wild images under real-world degradations. To produce real-world-like LQ-HQ image pairs, Real-ESRGAN (Wang et al., 2021) introduced a practicable datapipe that mixed multiple degradations and utilized adversarial learning to boost the model to recover images with more real details. APISR (Wang et al., 2024a) further expanded the degradation types with more image and video compressions. Recently, diffusion models (Rombach et al., 2022a;b; Esser et al., 2024) have surpassed the GAN in both text-to-image and image-to-image tasks. Following the trend, diffusion-based SR models (Wang et al., 2024b; Yu et al., 2024; Ai et al., 2025) proposed using controllable modules, *e.g.*, ControlNet Zhang et al. (2023a), to guide the diffusion model regenerating HQ images according to LQ condition. Then, (Wu et al., 2024b; Zhang et al., 2024; Yue et al., 2024a; Wang et al., 2025) distill the diffusion steps to improve throughout by examining the predicted latent with additional score networks (Wang et al., 2024e; Zhou et al., 2024) or discriminators (Sauer et al., 2024).

### 2.2 DIFFUSION DISTILLATION

To accelerate the diffusion model, existing distillation approaches reduce steps (Poole et al., 2022; Wang et al., 2024e; Zhou et al., 2024) and simplify the architecture (Kim et al., 2024; Hu et al., 2024;

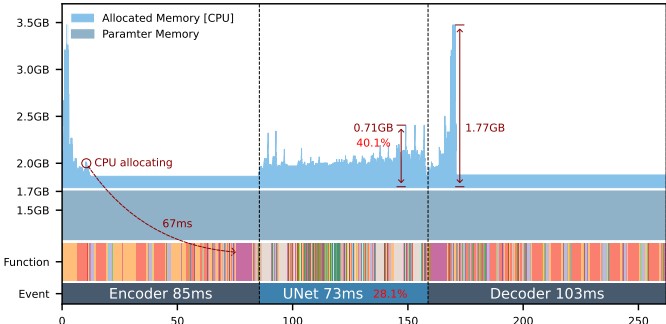

Figure 1: Profiling for GenDR with input size of $1024^2$px.

| Size | Time (ms) | | Mem (GB) | |
|---|---|---|---|---|
| | UNet | VAE | UNet | VAE |
| $512^2$ | 66 | 46 | 2.0 | 2.5 |
| $720^2$ | 66 | 101 | 2.2 | 2.8 |
| $1024^2$ | 73 | 191 | 2.7 | 3.5 |
| $1440^2$ | 187 | 419 | 3.0 | 5.1 |
| $2048^2$ | 534 | 1087 | 4.1 | 8.3 |
| $2880^2$ | 1710 | 3228 | 6.3 | 14.7 |
| $4096^2$ | 6201 | OOM | 11.1 | OOM |

Table 1: Comparison of varied input sizes. The memory is the maximum active memory after allocation for UNet or VAE.

Zhu et al., 2024). DreamFusion (Poole et al., 2022) introduced score distillation sampling (SDS), which leverages a pre-trained diffusion model to guide an arbitrary generator producing results satisfying the pre-trained distribution. ProlificDreamer (Wang et al., 2024e) and SiD (Zhou et al., 2024) developed SDS with variational score distillation (VSD) and identity transformation, which further improve stability and robustness. In the SR task, OSEDiff (Wu et al., 2024b) leveraged VSD and GenDR (Wang et al., 2025) developed CiD based on SiD to tailor a one-step SR model. For structural distillation, since disactivating parts of blocks or channels negligibly affects model performance, BK-SDM (Kim et al., 2024) and flux.1-lite-8B (Freepik, 2025) removed intermediate blocks of SD1.5 and FLUX and used feature distilling loss to train lighter version models. Diffusion2GAN (Kang et al., 2024) conducted knowledge distillation from diffusion to GAN via adversarial learning and LatentLPIPS. In the SR task, to lighten one-step diffusion, AdcSR (Chen et al., 2025a) performed knowledge distillation for a channel-pruned model.

## 3 MOTIVATION: REPLACING VAE WITH PIXEL(UN)SHUFFLE

**Efficiency bottleneck**. For the one-step diffusion model like GenDR (Wang et al., 2025), the diffusion process is conducted in the latent space, thereby necessitating a VAE to encode or decode latents. As shown in Fig. 1 and Tab. 1, we profiled the whole process with `torch.profiler` and visualized allocated GPU memory and latencies of different functions. We found that for the one-step model, VAE is the bottleneck for latency and GPU memory, which limits throughput, especially for high-resolution input. In detail, for a $2880^2$px image (*i.e.*, $720{\times}720$ LR input for $\times4$ SR), VAE induces additional growths of $1.82\times$ and $0.89\times$ for memory and latency, respectively. In practice, due to VAE and UNet trained under fixed small resolution (*e.g.*, $512{\times}512$) and GPU memory boundary, existing diffusion-based SR has to split the input image into multiple tiles, further stalling reasoning speed. To this end, modifying VAE is more profitable than modifying UNet.

**Fidelity bottleneck**. The existing VAEs conduct lossy compression that loses high-frequency details for input images. Even for the 16-channel VAE, the reconstruction results are far from reliable (Esser et al., 2024; Wang et al., 2025) (*e.g.*, PSNR is below 35 dB). For the SR task that chases pixel-wise fidelity, the VAE is the bottleneck restricting the reconstruction.

**Key and problem**. Based on the above findings, we pinpoint VAE as the bottleneck for both efficiency and fidelity, but *can we replace VAE with more efficient and loss-free functions? The answer is Yes!* We substitute the encoder with pixel-unshuffle and decoder with pixel-shuffle to reverse the latent-space diffusion processing into pixel-space. Despite intuitionistic, removing VAE faces two tricky problems:

- Pixel(un)shuffle with a large scale factor induces repeated pattern artifacts. Compared to AdcSR (Chen et al., 2025a) merely employing $\times2$ pixel-unshuffle to replace the encoder, we remove both encoder and decoder with $\times8$ pixel(un)shuffle to chase the ultimate efficiency in terms of both latency and memory occupancy. However, using pixelshuffle for $\times8$ upscaling is challenging since one inappropriate value of weight/bias in tail layers results in artifacts for all repeated $8{\times}8$ patches.

- Lack of a suitable discriminator for the pixelfuffled feature. Existing works (Chen et al., 2025a; Sauer et al., 2024) utilize pretrained diffusion models (UNet/DiT) as discriminators to adversarially

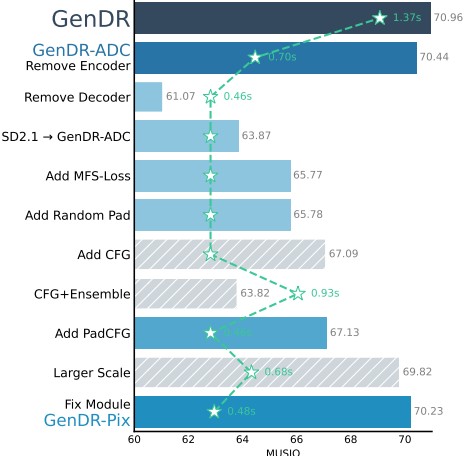

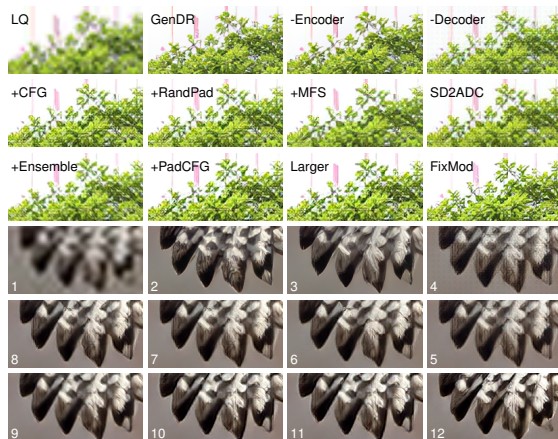

Figure 2: We remove the entire VAE for GenDR for higher efficiency. The bars are model MUSIQ score on RealLQ250. The hatched bar means the modification has not been adopted. The dotted curve shows runtime with 2560×1440 output.

Figure 3: We visualize the effect of our modifications step by step. The left-top corner shows the operation types corresponding to Fig. 2, and the left-down numbers are the operation order. The proposed strategies can effectively remove most artifacts.

distill the step/architecture by examining the generated latents. However, for pixel-unshuffled images, there exists no proper large model to guide distillation, further increasing the difficulty of reverse latent to pixel space.

**Relation to AdcSR**. The AdcSR (Chen et al., 2025a) has explored replacing the encoder with ×2 pixel-unshuffle and established an end-to-end ×4 SR model. However, there remain several limitations:

- AdcSR utilizes fixed ×2 pixel-unshuffle to encode images, which enforces the model to generate more pixels, limiting its extension in other upscaling factors or low-level tasks. We utilize the ×8 pixel-(un)shuffle to substitute VAE for flexibility and practicality.

- AdcSR only removes the encoder, while the decoder still costs more memory and latency than UNet. Moreover, removing the decoder results in more performance drops. This work studies the path to eliminating both the encoder and decoder for ideal efficiency.

- Compared to AdcSR with complex pruning and distillation strategies, this work only removes VAE and can achieve a comparable or even higher acceleration ratio.

## 4 METHODOLOGY

To overcome the above-mentioned issues, we propose a multi-stage adversarial distillation to fundamentally train a robust VAE-free model and a padding-based classifier-free guidance for stable inference in pixel space. In practice, we remove the VAE of GenDR to reverse the SR task to pixel space to propose **GenDR-Pix**. In Figs. 2 and 3, we present the roadmap from latent-based GenDR to pixel-based GenDR-Pix, which eliminates the entire VAE with slight performance drops.

### 4.1 MULTI-STAGE ADVERSARIAL DISTILLATION

To remove VAE and avoid repeated pattern artifacts, we propose multi-stage adversarial distillation schema. In Fig. 4, we exhibit an overview procedure for eliminating VAE. In Appendix A.1, we provide the detailed algorithm of the training schema.

**Stage I: Remove encoder**. We first replace the encoder with ×8 pixel-shuffle layer and utilize UNet from GenDR as the feature extractor of the discriminator. Given input image $\mathbf{x}_{lq}$, the teacher latents

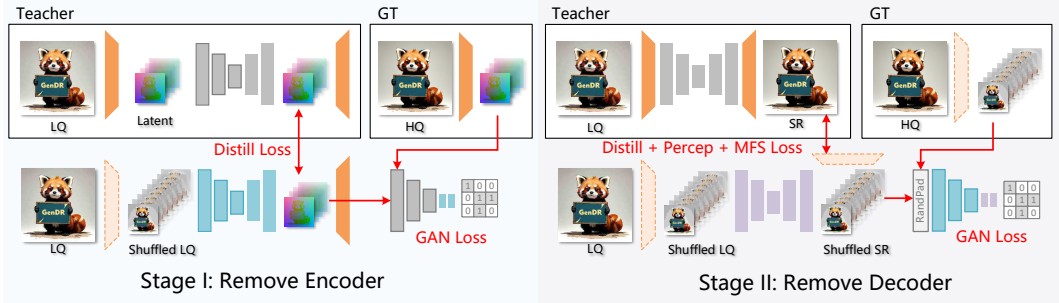

Figure 4: Illustration of training processes of **GenDR-Pix**. In Stage I, we replace the encoder with ×8 pixel-unshuffle and utilize half of the teacher (gray) to generate discriminative features, and blue blocks are trainable. In Stage II, we further remove the decoder with pixel-shuffle and utilize the Stage I model (blue), and the purple blocks are trainable.

$\mathbf{z}_{\text{tea}}$, HQ latents $\mathbf{z}_{\text{hq}}$, and discriminator $\mathcal{D}$, the overall training target for generator $\mathbf{z}_{\text{stu}} = \mathcal{G}_1(\mathbf{x}_{\text{lq}})$ is:

$$
\begin{aligned}
\mathcal{L}_{\mathcal{G}_1} &= ||\mathbf{z}_{\text{tea}} - \mathbf{z}_{\text{stu}}||_1 + \lambda_1 \cdot \text{softplus}(-\mathcal{D}(\mathbf{z}_{\text{stu}})), \\
\mathcal{L}_{\mathcal{D}} &= \text{softplus}(-\mathcal{D}(\mathbf{z}_{\text{hq}})) + \text{softplus}(\mathcal{D}(\mathbf{z}_{\text{stu}})).
\end{aligned}
\tag{1}
$$

In this stage, we obtain GenDR-Adc that maps low-quality pixels directly to high-quality latents.

**Stage II: Remove decoder**. Based on GenDR-Adc, we further replace the decoder with a pixel-shuffle layer. In practice, we find that using the Eq. (1) encounters severe performance drops due to the appearance of repeated pattern artifacts and the inappropriate latent-based discriminator.

To investigate and alleviate artifacts, we visualize several typical examples in Fig. 5. Specifically, we observe that the patterns are in the same repeated mode among all images with 8×8 patches. Moreover, we exhibit frequency-domain results where highlight points appear periodically, aligning with the pixel-shuffle scale. Thus, we identify that the distortion is caused by upscale, similar to artifacts raised by 2×2 embedding in FluxSR (Li et al., 2025). As the frequency characteristic of the artifacts is certain, we introduce a masked Fourier space (MFS) loss based on a band-rejection filter. In detail, given the SR/HQ image $\mathbf{y}_{\text{stu}}$ and $\mathbf{y}_{\text{tea}}$, we first calcu-

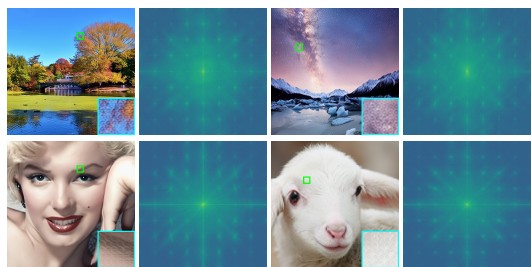

Figure 5: Illustration of artifacts in reconstructed image and their magnitude spectrum. (Zoom in for best view.)

late their amplitudes $|\mathcal{F}\{\mathbf{y}\}_{\mathbf{u},\mathbf{v}}|$ through FFT transformation. Then, we can compute the loss via the following formulation:

$$
\mathcal{L}_{\mathcal{F}} = ||\mathcal{M} \cdot (|\mathcal{F}\{\mathbf{y}_{\text{stu}}\}_{\mathbf{u},\mathbf{v}}| - |\mathcal{F}\{\mathbf{y}_{\text{tea}}\}_{\mathbf{u},\mathbf{v}}|)||_1,
\tag{2}
$$

where $\mathcal{M}$ is the mask inistialized from zero and satisfies $\mathcal{M}[:, i - s : i + s] = 1$ and $\mathcal{M}[j - s : j + s, :] = 1$ where $i \in \{0, \lfloor h/8 \rfloor, \ldots, h\}$ and $j \in \{0, \lfloor w/8 \rfloor, \ldots, w\}$, $2s$ is band width.

As to the discriminator, we utilize the first-stage generator $\mathcal{G}_1$ to evaluate the generated shuffled image since $\mathcal{G}_1$ natively uses pixel-unshuffle to encode inputs. Despite $\mathcal{G}_1$ being qualified for representation, as shown in Fig. 3, the detail generation still suffers performance degradation since the discriminator focuses on the discrete distribution embedded by a fixed pixel-unshuffle mode, which is susceptible to adversarial perturbations and easily induces model collapse. Inspired by E-LPIPS (Kettunen et al., 2019), we introduce random padding (RandPad) augmentation strategy to help the discriminator extract continuous representations for shuffled SR and HQ images. In detail, we randomly sample $p_h, p_w \in \{0, 1, \cdots, 7\}$ and execute padding for $\mathbf{y}_{\text{stu}}$ and $\mathbf{y}_{\text{hq}}$ before sending them to discriminator:

$$
\text{randpad}(\mathbf{y}) = \text{pad}(\mathbf{y}, [p_h, 8 - p_h, p_w, 8 - p_w]),
\tag{3}
$$

where $[p_h, 8 - p_h, p_w, 8 - p_w]$ are padding element numbers for left, right, top, and bottom sides.

Following previous work in pixel space (Kang et al., 2023; 2024), we add perceptual loss $\mathcal{L}_{\mathcal{P}}$ (Ding et al., 2020) to improve subjective fidelity. Given the input $\mathbf{x}_{\text{lq}}$, the teacher images $\mathbf{y}_{\text{tea}}$, the high-quality images $\mathbf{y}_{\text{hq}}$, we can optimize $\mathbf{y}_{\text{stu}} = \mathcal{G}_2(\mathbf{x}_{\text{lq}})$ by:

$$\mathcal{L}_{\mathcal{G}_2} = ||\mathbf{y}_{\text{tea}} - \mathbf{y}_{\text{stu}}||_1 + \lambda_1 \cdot \text{softplus}(-\mathcal{G}_1(\mathbf{y}_{\text{stu}})) + \lambda_2\mathcal{L}_{\mathcal{P}} + \lambda_3\mathcal{L}_{\mathcal{F}},$$
$$\mathcal{L}_{\mathcal{G}_1} = \text{softplus}(-\mathcal{G}_1(\text{randpad}(\mathbf{y}_{\text{hq}}))) + \text{softplus}(\mathcal{G}_1(\text{randpad}(\mathbf{y}_{\text{stu}}))).$$

(4)

In this stage, we reverse $\mathcal{G}_2$ in pixel space, which directly generates high-quality pixels.

## 4.2 PADDING CLASSIFIER-FREE GUIDANCE

To enable classifier-free (CFG) guidance in pixel space as well as erasing artifacts, we propose a padding-based classifier-free guidance (PadCFG) via incorporating self-ensemble and CFG strategies. Following Ho & Salimans (2022), we formulate the CFG prediction of GenDR-Pix $\mathcal{G}_2$ as:

$$\mathbf{y} = \omega \times \mathcal{G}_2(\mathbf{x}, \mathbf{c}_{\text{pos}}) + (1 - \omega) \times \mathcal{G}_2(\mathbf{x}, \mathbf{c}_{\text{neg}}),$$

(5)

where $\omega$ denotes the guidance scale. $\mathbf{c}_{\text{pos}}$ and $\mathbf{c}_{\text{neg}}$ are positive and negative condition, respectively. Since $\mathcal{G}_2(\mathbf{x})$ is a shuffled image, directly employing CFG in the pixel space will exacerbate artifacts, as shown in Fig. 3. A simple but effective solution is the self-ensemble strategy that combines restoration results from multiple augmented inputs, *e.g.*, rotation and flipping. Hence, we implement a self-ensemble by fusing results with various padding steps. Given the $n$ pairs padding setting $\{p_{h,i}, p_{w,i}\}$, the final ensembles result $\bar{\mathbf{y}}$ is calculated by:

$$\bar{\mathbf{y}} = \frac{1}{n}\sum_{i=1}^{n} \mathcal{G}_2(\text{pad}(\mathbf{x}, [p_{h,i}, 8 - p_{h,i}, p_{w,i}, 8 - p_{w,i}]))[p_{h,i} : h + p_{h,i}, p_{w,i} : w + p_{w,i}], \quad (6)$$

While effective in reducing artifacts, self-ensemble increases the inference batches and induces perceptual performance drops. To solve the problem, we empirically integrate Eqs. (5) and (6) by:

$$\bar{\mathbf{y}} = \omega \times \mathcal{G}_2(\text{pad}(\mathbf{x}, [4, 4, 4, 4]), \mathbf{c}_{\text{pos}}) + (1 - \omega) \times \mathcal{G}_2(\text{pad}(\mathbf{x}, [3, 5, 3, 5]), \mathbf{c}_{\text{neg}}). \quad (7)$$

To further improve visual quality in the inference phase, we try to enlarge the processing size and use an extra post-fixing module consisting of several convolution layers. More details and results can be found in Appendix A.1.

## 5 EXPERIMENTS

## 5.1 EXPERIMENTAL SETUP

**Datasets**. Following Wang et al. (2025), we adopt LSDIR (Li et al., 2023), FFHQ (Karras et al., 2019), DiffusionDB (Wang et al., 2022c), and select high-quality data from Laion (Schuhmann et al., 2022) to construct training set. To generate HQ-LQ image pairs, we use the Real-ESRGAN (Wang et al., 2021) and APISR (Wang et al., 2024a) degradation pipelines. We randomly crop $512 \times 512$ LQ/HQ images with dynamic scaling factor. During testing phase, we employ ImageNet-Test (Yue et al., 2024b), RealSR (Cai et al., 2019), RealSet80 (Yue et al., 2024b), and RealLR250 (Ai et al., 2025) to evaluate the effectiveness of the proposed GenDR-Pix under synthesised and real-world degradations.

**Metrics**. Following existing work (Wang et al., 2025; Yue et al., 2024a), we leverage several full-reference and non-reference IQA metrics to evaluate restoration quality, including PSNR, SSIM (Wang et al., 2004), LPIPS (Zhang et al., 2018a), NIQE (Mittal et al., 2012), LIQE (Zhang et al., 2023b), CLIPIQA (Wang et al., 2023), MUSIQ (Ke et al., 2021), Q-Align (Quality) (Wu et al., 2024a), and DeQA (You et al., 2025).

**Implementation details**. GenDR-Pix is based on GenDR (Wang et al., 2025), which employs SD2.1 UNet (Rombach et al., 2022b) and vae-kl-f8-d16 [1]. To eliminate VAE, we reinitialized the 16-channel head and tail convolutions of GenDR with 192-channel ones in stages I and II, respectively. For the discriminator, we used frozen GenDR and GenDR-Adc as the feature extractor module and extra learnable MLP heads as LADD (Sauer et al., 2024). During the training procedure, we use the default

---

[1]https://huggingface.co/ostris/vae-kl-f8-d16

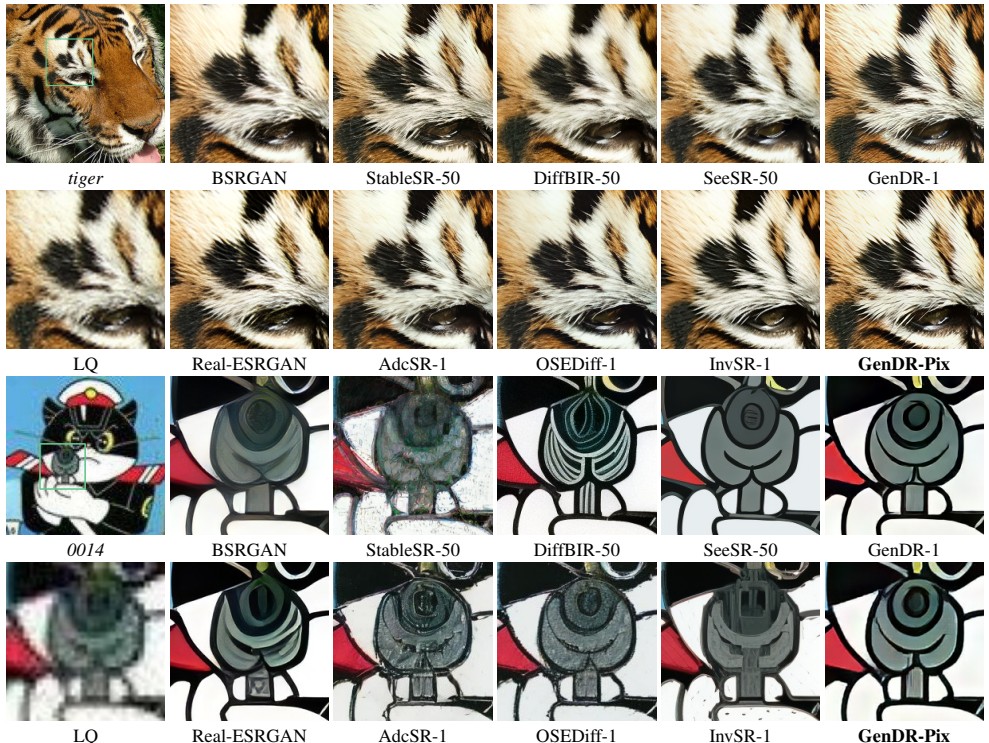

Figure 6: Visual comparison of GenDR-Pix with other methods for ×4 task on RealSet80 dataset.

AdamW (Loshchilov & Hutter, 2018) optimizer and a constant learning rate $1e^{-5}$ under BFloat16 precision. The loss hyperparameters $\lambda_{1,2,3}$ are 0.05, 1, 0.1, respectively. The batch and patch sizes are 512 and 512×512. All models are trained on 8 NVIDIA A100 GPUs and the PyTorch (Paszke et al., 2019) framework with the help of deepspeed ZeRO2 (Rajbhandari et al., 2020). During the inference phase, we set PadCFG 1.8 for better restoration quality.

## 5.2 COMPARISION WITH STATE-OF-THE-ARTS

To examine the restoration and inference performance of the proposed GenDR-Pix, we compare it with multiple models, including 1) **GAN**: BSRGAN (Zhang et al., 2021), Real-ESRGAN (Wang et al., 2021), and RealESRHAT (Chen et al., 2023); 2) **one-step diffusion**: SinSR (Wang et al., 2024d), OSEDiff (Wu et al., 2024b), InvSR (Yue et al., 2024a), AdcSR (Chen et al., 2025a), and GenDR (Wang et al., 2025); 3) **multi-step diffusion**: StableSR (Wang et al., 2024b), DiffBIR (Luo et al., 2023), SeeSR (Wu et al., 2024c), and DreamClear (Ai et al., 2025).

**Quantitative comparison**. In Tabs. 2 and 3, we compare the proposed GenDR-Pix with existing SOTA methods in terms of both restoration quality and inference performance. Generally, GenDR is the fastest model among all methods, even surpassing GAN-based models. Compared to OSEDiff and SeeSR, GenDR-Pix gains acceleration of 3.2× and 198.7×, respectively. Moreover, GenDR-Pix stands as the runner-up for both FR-IQA and NR-IQA metrics. In detail, GenDR-Pix advances OSEDiff by 0.6dB (PSNR) and 0.04 (CLIPIQA) on ImageNet-Test. Compared to the original GenDR, GenDR-Pix improves the PSNR/SSIM by a large margin while slightly decreasing perceptual quality. On RealSet80 benchmark, GenDR-Pix exhibits competitive performance with OSEDiff and InvSR. In Appendix A.2.2, we extend more results on real-world image datasets, including **4K benchmark**.

**Qualitative comparison**. In Fig. 6, we depict the visual comparison of GenDR against other approaches. GenDR-Pix obtains comparable results to GenDR and surpasses other approaches. Specifically, our model faithfully restores the tiger hair with more delicacy. For image *0014*, GenDR-Pix removes the heavy compression artifact and maintains the semantical structure.

Table 2: Quantitative comparison (average Parameters, MACs, and IQA metrics) on ImageNet-Test. The sampling step number is marked in the format of "Method name-Steps" for diffusion-based methods. The efficiency metrics are calculated with 512×512 input on NVIDIA A100 GPUs. The best results for all methods are highlighted in **bold** and underlined, while the best *one-step* diffusion methods are reported in red and blue.

| Methods | #Params↓ | #MACs↓ | Metrics | | | | | | |
|---|---|---|---|---|---|---|---|---|---|
| | | | PSNR↑ | SSIM↑ | LPIPS↓ | NIQE↓ | CLIPIQA↑ | MUSIQ↑ | Q-Align↑ |
| BSRGAN | 16.70M | 293G | 27.05 | 0.7453 | 0.2437 | 4.5345 | 0.5703 | 67.72 | 3.6057 |
| Real-ESRGAN | 16.70M | 293G | 26.62 | 0.7523 | 0.2303 | 4.4909 | 0.5090 | 64.81 | 3.4230 |
| Real-HATGAN | 20.77M | 417G | **27.15** | **0.7690** | **0.2044** | 4.7834 | 0.4594 | 63.43 | 3.3244 |
| StableSR-50 | 1410M | 79940G | 26.00 | 0.7317 | 0.2327 | 4.9378 | 0.5768 | 64.54 | 3.4378 |
| DiffBIR-50 | 1717M | 24234G | 25.45 | 0.6651 | 0.2876 | 4.9289 | 0.7486 | 73.04 | 4.3228 |
| SeeSR-50 | 2524M | 65857G | 25.73 | 0.7072 | 0.2467 | 4.3530 | 0.6981 | 72.25 | 4.2412 |
| ⋆DreamClear-50 | 2212M | - | 24.76 | 0.6672 | 0.2463 | 5.3787 | **0.7646** | 70.08 | 4.0919 |
| SinSR-1 | 119M | 2649 | 26.98 | 0.7308 | 0.2288 | 5.2506 | 0.6607 | 67.70 | 3.8090 |
| OSEDiff-1 | 1775M | 2265G | 24.82 | 0.7017 | 0.2431 | 4.2786 | 0.6778 | 71.74 | 4.0674 |
| AdcSR-1 | 456M | 496G | 25.07 | 0.6982 | 0.2432 | 4.1947 | 0.7055 | 72.21 | 4.1382 |
| InvSR-1 | 1298M | 1820G | 23.81 | 0.6777 | 0.2547 | 4.3935 | 0.7114 | 72.38 | 3.9867 |
| GenDR-1 | 933M | 1637G | 24.14 | 0.6878 | 0.2652 | **4.1336** | 0.7395 | **74.68** | **4.3612** |
| **GenDR-Pix** | 866M | 344G/744G | 25.49 | 0.7286 | 0.2252 | 4.1892 | 0.7168 | 72.85 | 4.2082 |

⋆: inference using 3 A100 GPU to run an image.

Table 3: Quantitative comparison (average Inference time and IQA metrics) on RealSet80.

| Methods | #Runtime↓ | Metrics | | | | | | |
|---|---|---|---|---|---|---|---|---|
| | | NIQE↓ | PI↓ | LIQE↑ | CLIPIQA↑ | MUSIQ↑ | Q-Align↑ | DeQA↑ |
| BSRGAN | 36ms | 4.4428 | 4.0276 | 3.8449 | 0.6262 | 66.63 | 4.1207 | 3.8825 |
| Real-ESRGAN | 36ms | 4.1517 | 3.8843 | 3.7392 | 0.6190 | 64.49 | 4.1696 | 3.8906 |
| Real-HATGAN | 116ms | 4.4705 | 3.8843 | 3.4927 | 0.5502 | 63.21 | 4.1077 | 3.8444 |
| StableSR-50 | 3731ms | **3.3999** | **3.0314** | 3.8516 | 0.7399 | 67.58 | 4.0870 | 3.8787 |
| DiffBIR-50 | 6213ms | 5.1389 | 3.9544 | 4.0472 | **0.7404** | 68.72 | 4.3206 | 4.0668 |
| SeeSR-50 | 6359ms | 4.3749 | 3.7454 | 4.2797 | 0.7124 | 69.74 | 4.3056 | 4.0728 |
| ⋆DreamClear-50 | 6892ms | 3.7257 | 3.4157 | 3.9628 | 0.7242 | 67.22 | 4.1206 | 3.9429 |
| SinSR-1 | 120ms | 5.6103 | 4.2697 | 3.5957 | 0.6634 | 63.79 | 4.0954 | 3.9713 |
| OSEDiff-1 | 103ms | 3.9763 | 3.6894 | 4.1298 | 0.7037 | 69.19 | 4.3057 | **4.0916** |
| AdcSR-1 | 33ms | 4.0005 | 3.5688 | 4.2313 | 0.7079 | 69.73 | 4.2298 | 4.0530 |
| InvSR-1 | 115ms | 4.0266 | 3.4524 | 4.2906 | 0.7271 | 69.79 | 4.3014 | 4.0045 |
| **GenDR-Pix** | 32ms | 4.1010 | 3.6709 | **4.3078** | 0.7190 | **69.92** | 4.2891 | 4.0747 |

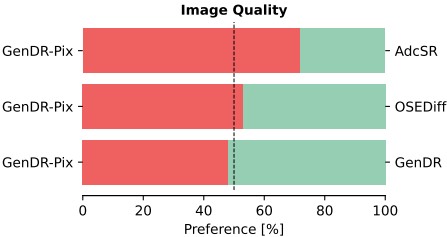

Figure 7: User studies on image quality.

**User study**. We conducted a user study to evaluate our model's restoration quality. Users were asked to select the higher-quality image from two images, one restored by GenDR-Pix and the other restored by AdcSR, OSEDiff, or GenDR. As shown in Fig. 7, our model received comparable votes as the original GenDR and OSEDiff, and led AdcSR by quarter, demonstrating the effectiveness of the proposed distillation strategy to preserve the ability to produce high-quality images.

## 5.3 Ablation Studies

**Effect of removing VAE**. In Tab. 6, we compare the original GenDR (Wang et al., 2025), GenDR-Adc (Chen et al., 2025a) (Removing Encoder), and GenDR-Pix (Removing VAE) in terms of both quality and runtime performance. Compared to the GenDR baseline, the proposed GenDR-Pix maintains the restoration quality but saves approximately 65% time, 61% GPU memory, and 55% computations. Moreover, if we allow for a slight performance drop, **the runtime for 4K image can be reduced to less than 1 second** by disabling CFG. The Fig. 9 shows the visual comparison in which GenDR-Pix can maintain detail and sharpness as the baseline. We also provide extended performance comparison in Fig. 8, where GenDR-Pix remarkably improves the throughput and lowers the maximum memory employment. Moreover, GenDR is the only model supporting SR to 8K, dispensing with the cropping strategy. The results demonstrate the effectiveness of removing VAE to improve efficiency.

Table 6: Quantitative comparison between GenDR, GenDR-Adc, and GenDR-Pix on RealLR250. The inference performances are tested with 4K (3840×2160) output on A100 GPU, except for #MACs tested with 512×512 output. "⋆" represents model without PadCFG strategy.

| Model | VAE | #Runtime | #Memory | #MACs | #Params | MUSIQ | CLIPIQA |
|---|---|---|---|---|---|---|---|
| GenDR | VAE | 4.92s | 20.75GB | 1637G | 933M | 70.96 | 0.6891 |
| GenDR-Adc | Remove Encoder | 2.69s (-45.3%) | 17.75GB (-14.5%) | 1097G (-33.0%) | 894M | 70.44 | 0.6714 |
| **GenDR-Pix** | Remove VAE | 1.75s (-64.5%) | 8.01GB (-61.4%) | 744G (-54.6%) | 866M | 70.23 | 0.6843 |
| **GenDR-Pix⋆** | Remove VAE | 0.87s (-82.3%) | 5.03GB (-75.8%) | 344G (-78.9%) | 866M | 68.64 | 0.6615 |

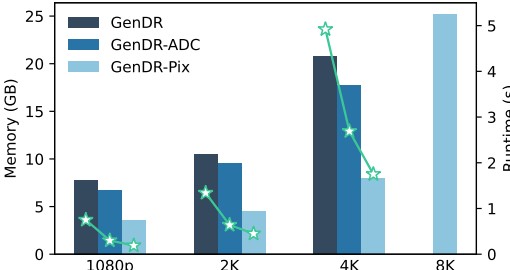

Figure 8: Performance evaluation (1080p to 8K).

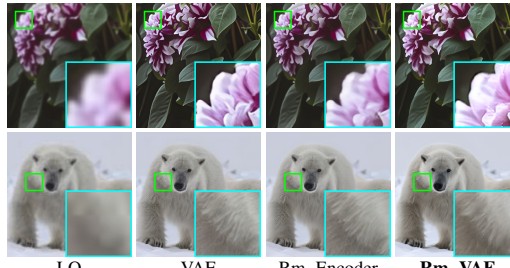

Figure 9: Visual comparison of varied VAE set.

| Method | Fomulation | #Runtime | LPIPS | MUSIQ | CLIPIQA |
|---|---|---|---|---|---|
| w/o CFG | $\mathbf{p}$ | 0.87s | 0.2391 | 58.23 | 0.4924 |
| CFG | $\omega \times \mathbf{p} + (1-\omega) \times \mathbf{n}$ | 1.73s | 0.2425 | 58.67 | 0.5005 |
| CFG + Ensemble | $\frac{1}{n}\sum\left[\omega \times \mathbf{p}_{(p_h,p_w)} + (1-\omega) \times \mathbf{n}_{(p_h,p_w)}\right]$ | 3.54s | 0.2273 | 56.31 | 0.4800 |
| PadCFG | $\omega \times \mathbf{p}_{(4,4)} + (1-\omega) \times \mathbf{n}_{(1,1)}$ | 1.75s | 0.2575 | 58.13 | 0.5119 |
| PadCFG | $\omega \times \mathbf{p}_{(4,4)} + (1-\omega) \times \mathbf{n}_{(2,2)}$ | 1.75s | 0.2474 | 58.85 | 0.5161 |
| PadCFG | $\omega \times \mathbf{p}_{(4,4)} + (1-\omega) \times \mathbf{n}_{(3,3)}$ | 1.75s | 0.2413 | 59.12 | 0.5066 |

Table 7: Ablation study of PadCFG with varied pad settings. $\{\mathbf{p}, \mathbf{n}\}$ represents results with positive and negative prompts and the subscripts $(p_h, p_w)$ are padding hyparameters for each side.

**Effect of Multi-Stage Adversarial Distillation**. Since training GenDR-Pix within one stage is unavailable and AdcSR has discussed similar results of Stage I, we conduct ablation studies for Stage II.

In detail, we validate the effectiveness of discriminator, RandPad augmentation, and MFS loss. We first examine the discriminator in Tab. 4, in which the disablement of adversarial learning degrades the perceptual quality by 0.12 and 0.0533 in MUSIQ and CLIPIQA. Compared to our schema using GenDR-Adc for adversarial feature, directly using GenDR will induce 2.20 (MUSIQ) and 0.0307 (CLIPIQA) drops. Employing RandPand improves CLIPIQA by 0.0191 and decreases NIQE by 0.17. Tab. 5 discusses the impact of MFC loss, where omitting frequency loss leads to huge performance drops among all IQA metrics.

Table 4: Ablation study on discriminator.

| Discriminator | RandPad | NIQE | MUSIQ | CLIPIQA |
|---|---|---|---|---|
| w/o Dis. | - | 4.8343 | 60.95 | 0.4924 |
| GenDR | - | 6.0660 | 61.07 | 0.5457 |
| GenDR-Adc | ✗ | 4.8380 | 63.87 | 0.5764 |
| GenDR-Adc | ✓ | 4.6648 | 63.89 | 0.5937 |

Table 5: Ablation study on MFC Loss.

| Loss | NIQE | LIQE | MUSIQ | CLIPIQA |
|---|---|---|---|---|
| w/o Freq loss | 4.8380 | 2.9055 | 63.87 | 0.5764 |
| MFC loss | 4.1121 | 3.2062 | 65.77 | 0.5700 |

**Effect of PadCFG**. To evaluate PadCFG, we compare several variants and different settings in Tab. 7, disabling other optimizations for fairness. The proposed Eq. (7) maintains latency and fidelity as vanilla CFG but advances 0.45 and 0.0061 in MUSIQ and CLIPIQ, achieving better trade-offs between fidelity and perceptual quality. We also validate the guidance scales in Appendix A.2.1.

## 6 CONCLUSION

This work introduces a guideline to remove the entire VAE for diffusion-based SR. In detail, we replace the encoder and decoder with ×8 pixel-unshuffle and pixel-shuffle, respectively. To alleviate the artifacts brought by pixel(un)shuffle, we introduce a multi-stage adversarial distillation for training and PadCFG for inference. Based on GenDR and proposed strategies, we obtain GenDR-Pix, which accelerates by 2.8× and saves 60% GPU memory with negligible quality degradation.

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

# A  APPENDIX

## A.1  ALOGRORITHM DETAILS

**Overall scheme**. In Algorithm 1, we summarize the training schema of the proposed GenDR-Pix. We first remove the encoder and then remove the decoder via adversarial learning. The $\mathcal{L}_{\mathcal{F}}$ is calculated according to Eq. (2), and we utilize DISTS (Ding et al., 2020) as $\mathcal{L}_{\mathcal{P}}$ to estimate the perceptual similarity between the reference teacher results and student results.

---

**Algorithm 1:** Training Scheme of GenDR-Pix

---

**Hyper-patameters:** $\lambda_1 = 0.05$, $\lambda_2 = 1$, $\lambda_3 = 0.1$, $t = 499$

▷ *Stage I: Remove Encoder*

**Input:** Pretrained GenDR network $\mathcal{G}_\mu$, GenDR-Adc $\mathcal{G}_\theta$, discriminative head $h_\phi$
**Initialization** $\theta \leftarrow \mu$, initialize $\phi$
**repeat**
 Sample HR and LR image pairs $(\mathbf{x}_h, \mathbf{x}_l)$ and prompt $c$ from batch $\mathcal{B}$; calculate the latent $\mathbf{z}_h$ and $\mathbf{z}_l$; let $\mathbf{z}_g = \mathcal{G}_\theta(\mathbf{x}_l; t, c)$ and reference model generate $\mathbf{z}_r = \mathcal{G}_\mu(\mathbf{z}_l; t, c)$; then calculate adversarial features $\mathbf{d}_g = \mathcal{G}_\mu(\mathbf{z}_g; t, c)$ and $\mathbf{d}_h = \mathcal{G}_\mu(\mathbf{z}_h; t, c)$; $\mathrm{sg}[\cdot]$ and $\mathrm{sp}[\cdot]$ are stop gradient and softplus operations
 Update $\phi$ with $\phi = \phi - \eta \nabla_\phi \mathcal{L}_\phi$, where

$$\mathcal{L}_\phi = \mathrm{sp}[-h_\phi(\mathrm{sg}[\mathbf{d}_h])] + \mathrm{sp}[h_\phi(\mathrm{sg}[\mathbf{d}_g])]$$

 Update $\mathcal{G}_\theta^1$ with $\theta = \theta - \eta \nabla_\theta \mathcal{L}_\theta$, where

$$\mathcal{L}_\theta = ||\mathbf{z}_r - \mathbf{z}_g||_1 + \lambda_1 \cdot \mathrm{sp}[-h_{\mathrm{sg}[\phi]}(\mathbf{z}_g)]$$

**until** achieving convergence or processing 20M image
**Output:** $\mathcal{G}_\theta$

▷ *Stage II: Remove Decoder*

**Input:** Pretrained GenDR network $\mathcal{G}_\mu$, GenDR-Adc $\mathcal{G}_\theta$, GenDR-Pix $\mathcal{G}_\psi$, discrimimative head $h_\phi$
**Initialization** $\psi \leftarrow \theta$, initialize $\phi$
**repeat**
 Sample HR and LR image pairs $(\mathbf{x}_h, \mathbf{x}_l)$, and prompt $c$ from batch $\mathcal{B}$; calculate the latent $\mathbf{z}_l$; let $\mathbf{x}_g = \mathcal{G}_\psi(\mathbf{x}_l; t, c)$ and reference model generate $\mathbf{z}_r = \mathcal{G}_\mu(\mathbf{z}_l; t, c)$; decode $\mathbf{z}_r$ to image $\mathbf{x}_r$, then calculate adversarial features $\mathbf{d}_g = \mathcal{G}_\theta(\mathbf{x}_g; t, c)$ and $\mathbf{d}_h = \mathcal{G}_\theta(\mathbf{x}_h; t, c)$; RandPad augument $\hat{\mathbf{d}}_h$ and $\hat{\mathbf{d}}_g$
 Update $\phi$ with $\phi = \phi - \eta \nabla_\phi \mathcal{L}_\phi$, where

$$\mathcal{L}_\phi = \mathrm{sp}[-h_\phi(\mathrm{sg}[\hat{\mathbf{d}}_h])] + \mathrm{sp}[h_\phi(\mathrm{sg}[\hat{\mathbf{d}}_g])]$$

 Update $\mathcal{G}_\psi$ with $\psi = \psi - \eta \nabla_\psi \mathcal{L}_\psi$, where

$$\mathcal{L}_\psi = ||\mathbf{x}_r - \mathbf{x}_g||_1 + \lambda_1 \cdot \mathrm{sp}[-h_{\mathrm{sg}[\phi]}(\mathbf{x}_g)] + \lambda_2 \mathcal{L}_{\mathcal{P}}(\mathbf{x}_r, \mathbf{x}_g) + \lambda_3 \mathcal{L}_{\mathcal{F}}(\mathbf{x}_r, \mathbf{x}_g)$$

**until** achieving convergence or processing 20M image
**Output:** $\mathcal{G}_\psi$

---

**RandPad**. As we also employ RandPad to augment the discriminative features, we provide its pseudocode in Fig. 10. The padding dimensions `pad_h` and `pad_w` are randomly chosen. We use the 'reflect' mode to perform padding on input images to avoid boundary misalignment of `image_neg` and `image_pos`.

**PadCFG**. In Fig. 11, we present the pseudocode of PadCFG. Based on Eq. (7), we implement the PadCFG in the same manner as the vanilla CFG, yet we introduce extra padding operations before concatenation. Since the computational complexity of the padding operation is almost negligible, our method can earn a "free lunch" for qualitative improvement.

**FixMod**. We realize a simple post-fix module (FixMod) to avoid the aberration and further reduce the influence of artifacts. Precisely, we construct it by combining the wavelet color fix and a small pretrained UNet. We also conduct jointly training of FixMod and GenDR-Pix to improve robustness and generatlity.

```
# RandPad for GT image
pad_h = random.randint(0, 8)
pad_w = random.randint(0, 8)

with no_grad():
    gt_image = pad(gt_image, (pad_h, 8-pad_h, pad_w, 8-pad_w),
        mode='reflect')
    # encode with pixel-unshuffle
    gt_latents = pixelunshuffle(gt_image, 8)
```

Figure 10: Pseudocode for RandPad.

```
image_neg = pad(image, (4, 4, 4, 4), mode='reflect')
image_pos = pad(image, (3, 5, 3, 5), mode='reflect')

image_input = concat([image_neg, image_pos], dim=0)

# encode with pixel-unshuffle
latents = pixelunshuffle(image_input, 8)

pred = unet(latents, t, prompt)
pred_neg, pred_pos = pred.chunk(2, dim=0)

# PadCFG
latents = pred_neg + guidence_scale*(pred_pos - pred_neg)

# encode with pixel-shuffle
image_output = pixelshuffle(latents, 8)[:,:,4:-4,4:-4]
```

Figure 11: Pseudocode for PadCFG.

## A.2 ADDITIONAL RESULTS FOR GENDR-PIX

### A.2.1 ABLATION STUDIES

**Effects of guidance scale**. Similar to vanilla CFG, the proposed PadCFG can balance the detail diversity and fidelity of generated samples by adjusting the guidance scale. In Fig. 13, we test varied guidance scales from 1.0 to 2.0 and compute PSNR and MUSIQ. Specifically, as PadCFG guidance increases, PSNR decreases while MUSIQ (perceptual quality) improves, indicating an inverse relationship where higher guidance enhances perceptual quality (as measured by MUSIQ) at the cost of PSNR reduction. In Fig. 12, we compare visual results under different guidance scales. Increasing the guidance scale leads to enhanced sharpness and more intricate details.

**Effects of FixMod**. In Tab. 8, we conduct an ablation study on the effects of FixMod. Overall, FixMod significantly improves perceptual scores by 3.15 and 0.0097 on MUSIQ and CLIP-IQA, respectively, outperforming processed images at larger resolutions. Additionally, FixMod is a lightweight module, introducing only 0.02 seconds of latency increase, as demonstrated in

Table 8: Ablation study on FixMod.

| Method | Color Fix | ARNIQA | MUSIQ | CLIPIQA |
|---|---|---|---|---|
| Baseline | - | 0.6700 | 67.13 | 0.6013 |
| Large Res. | - | 0.6854 | 69.81 | 0.6843 |
| FixMod | ✗ | 0.6935 | 70.23 | 0.6975 |
| FixMod | ✓ | 0.7033 | 70.28 | 0.7010 |

Fig. 2. In Fig. 14, we visualize the results of varied post-processing configurations. Notably, while post-processing introduces metric improvement, it provides no additional details but alleviates certain artifacts in the outputs.

**Effects of dataset scale and training period**. During the training process, we utilize approximately 20 million LQ-HQ image pairs under a longer training scheduler to optimize GenDR-pix, which surpasses most existing one-step models. In summary, we employ the data and scheduler strategy for two reasons: 1) GenDR-Pix depends on full-parameter training to transfer from latent space to pixel

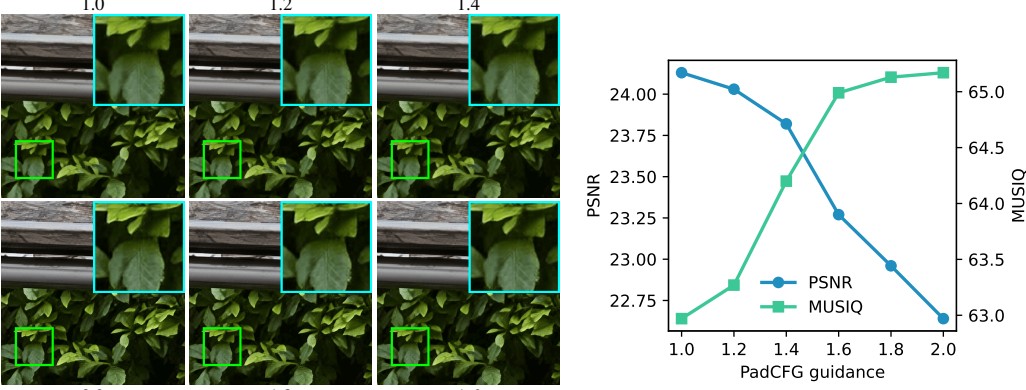

Figure 12: Visual comparison of varied PacCFG guidance scales on "Nikon_005_LR4" from RealSR.

Figure 13: Impact of guidance scale.

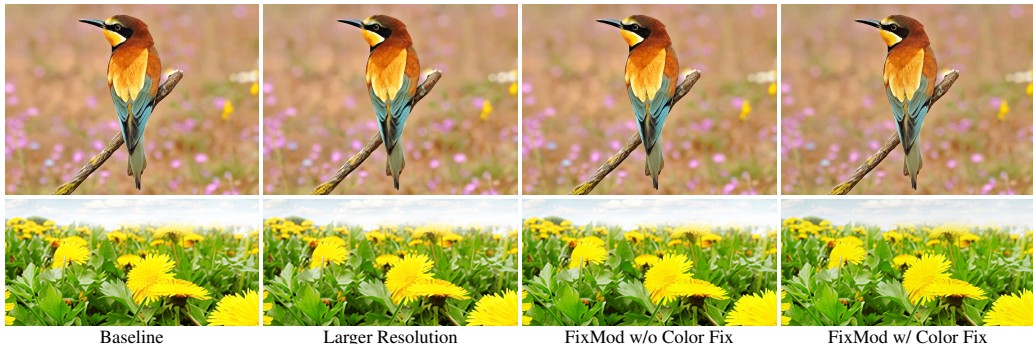

Figure 14: Visual comparison between baseline, larger resolution processing, and FixMod.

space, which requires more data and time to stabilize the training process and avoid overfitting; 2) GenDR-Pix is trained under more complex degradation, which integrates multiple upscale factors using both the RealESRGAN pipeline and the APISR pipeline, requiring more data for comprehensive distillation.

To directly quantify the influence of dataset scale and training steps, we visualize performance trajectories in Fig. 15. All three curves show non-linear "phase-transition" behavior. Specifically, ClipIQA experiences a sharp improvement around 50k–100k iterations. However, more importantly, the absolute performance consistently increases across all regimes, even before these transition points. This indicates that the observed gains are not accidental artifacts of data scaling, but a stable consequence of the proposed distillation strategy. Moreover, the performance gap between 1M and 20M images is only 1.15 on MUSIQ, which is significantly smaller than the influence of our distillation strategy.

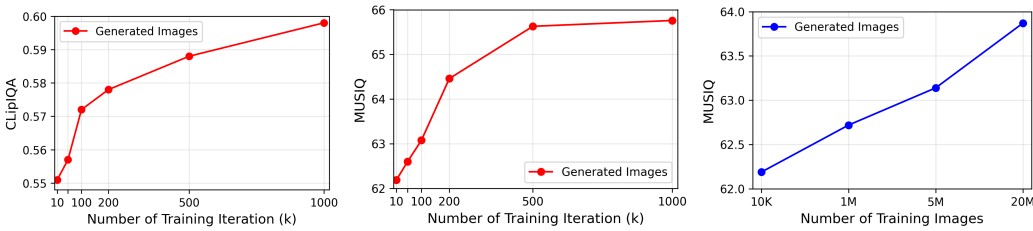

Figure 15: Impact of training iteration and data.

Table 9: Quantitative comparison (average Parameters, time, and IQA metrics) on RealSR (×4).

| Methods | #Runtime↓ | Metrics | | | | | | |
|---|---|---|---|---|---|---|---|---|
| | | PSNR↑ | SSIM↑ | LPIPS↓ | NIQE↓ | CLIPIQA↑ | MUSIQ↑ | Q-Align↑ |
| BSRGAN | 36ms | **26.51** | 0.7746 | 0.2685 | 4.6501 | 0.5439 | 63.59 | 3.8277 |
| Real-ESRGAN | 36ms | 25.85 | 0.7734 | 0.2729 | 4.6788 | 0.4901 | 59.69 | 3.9185 |
| Real-HATGAN | 116ms | 26.22 | **0.7894** | **0.2409** | 5.1189 | 0.4336 | 58.41 | 3.8353 |
| StableSR-50 | 3731ms | 24.52 | 0.6733 | 0.3658 | **3.4665** | 0.6897 | 66.87 | 3.9862 |
| DiffBIR-50 | 6213ms | 26.28 | 0.7251 | 0.3187 | 5.8009 | 0.6743 | 64.28 | 3.9182 |
| SeeSR-50 | 6359ms | 26.19 | 0.7555 | 0.2809 | 4.5366 | 0.6826 | 66.31 | 3.9862 |
| *DreamClear-50 | 6892ms | 24.14 | 0.6963 | 0.3155 | 3.9661 | 0.6730 | 63.74 | 3.9705 |
| SinSR-1 | 120ms | 25.99 | 0.7072 | 0.4022 | 6.2412 | 0.6670 | 59.22 | 3.8800 |
| OSEDiff-1 | 103ms | 24.57 | 0.7202 | 0.3036 | 4.3408 | 0.6829 | 67.30 | 4.0664 |
| AdcSR-1 | 33ms | 25.69 | 0.7362 | 0.2825 | 4.1846 | 0.6695 | 67.47 | 4.0772 |
| InvSR-1 | 115ms | 24.50 | 0.7262 | 0.2872 | 4.2218 | 0.6919 | 67.47 | 4.2085 |
| GenDR-1 | 77ms | 23.18 | 0.7135 | 0.2859 | 4.1588 | **0.7014** | **68.36** | **4.2388** |
| **GenDR-Pix** | 32ms | 25.03 | 0.7513 | 0.2702 | 4.0751 | 0.6816 | 66.29 | 4.1815 |

⋆: inference using 3 A100 GPU to run an image.

Table 10: Quantitative comparison (average Parameters, time, and IQA metrics) on RealSR and RealLQ250 with scal factor×2 and ×3.

| Methods | Scale | RealSR | | | | RealLQ250 | | |
|---|---|---|---|---|---|---|---|---|
| | | PSNR↑ | LPIPS↓ | ClipIQA↑ | MUSIQ↑ | NIQE↓ | CLIPIQA↑ | MUSIQ↑ |
| Real-ESRGAN | ×2 | 28.02 | 0.2062 | 0.6014 | 63.54 | 5.5589 | 0.6883 | 67.72 |
| OSEDiff-1 | ×2 | 23.29 | 0.2774 | 0.7214 | 67.56 | 4.1813 | 0.7056 | 71.16 |
| GenDR-1 | ×2 | 27.18 | 0.1753 | 0.6152 | 66.82 | 4.3286 | 0.6981 | 70.21 |
| **GenDR-Pix** | ×2 | 27.95 | 0.1660 | 0.6797 | 65.14 | 4.1320 | 0.7194 | 71.59 |
| Real-ESRGAN | ×3 | 26.69 | 0.2475 | 0.5762 | 62.14 | 4.4329 | 0.6232 | 65.90 |
| OSEDiff-1 | ×3 | 20.35 | 0.3165 | 0.6183 | 66.97 | 3.9106 | 0.6969 | 70.64 |
| GenDR-1 | ×3 | 21.90 | 0.2416 | 0.5841 | 64.61 | 3.8648 | 0.6781 | 71.39 |
| **GenDR-Pix** | ×3 | 26.80 | 0.2065 | 0.6128 | 60.96 | 3.8545 | 0.7067 | 71.43 |

### A.2.2 QUANTITATIVE RESULTS

In Tab. 9, we present additional results for RealSR (Cai et al., 2019). Specifically, GenDR-Pix achieves higher SSIM and LPIPS values than any other one-step diffusion method, benefiting from lossless pixelshuffle and pixelunshuffle operations. Regarding NR-IQA, GenDR-Pix achieves a score of 4.18, surpassing OSEDiff by 0.1150. Specifically, GenDR-Pix can compete with some multi-step diffusion models like DiffBIR by accelerating by 194.2×. In Tab. 10, we compare the proposed GenDR-Pix with RealESRGAN, OSEDiff, and GenDR under multiple scale factor. Unlike AdcSR tailored for ×4 SR, the proposed method achieves remarkable performance for both ×2 and ×3 SR, showing its effectiveness and robustness.

Moreover, we examine GenDR-Pix with the real-world SR for 4K images on RealDeg (Chen et al., 2024) benchmarks. Since most methods need to process 4K SR with a tiling strategy, which is extremely slow, we only include GenDR as a representative latent-based method during testing. Tab. 11 presents the efficiency and restoration comparison of proposed GenDR-Pix with BSRGAN (Zhang et al., 2021), AdcSR (Chen et al., 2025a), and GenDR (Wang et al., 2025). Specifically, only BSRGAN and GenDR-Pix can process 4K images without failure and tile strategy. In general, GenDR-Pix is slightly behind GenDR but surpasses the other method by a large margin.

Notably, we observe that the performance gap between GenDR and GenDR-Pix slightly widens, which may be attributed to the overlap introduced by the tiling strategy. For efficiency performance, compared to AdcSR and GenDR, GenDR-Pix performs 4.29× and 12.99× acceleration, respectively. Compared to BSRGAN and AdcSR, GenDR saves 49.1% and 64.8% memory consumption.

### A.2.3 QUALITATIVE RESULTS

In Fig. 16, we depict the visual comparison of GenDR against other approaches. GenDR-Pix obtains comparable results to GenDR and surpasses other approaches. Specifically, GenDR-Pix faithfully

Table 11: Quantitative comparison (average Parameters, Inference time, and IQA metrics) on Social Media subset from RealDeg. The runtime and memory are measured on RealDeg with an A100 GPU.

| Methods | Status | #Runtime↓ | #Memory↓ | Metrics | | | | |
|---|---|---|---|---|---|---|---|---|
| | | | | NIQE↓ | LIQE↑ | ARNIQA↑ | MANIQA↑ | CLIPIQA↑ |
| BSRGAN | Pass | 2.10s | 29.65GB | 4.1816 | 2.9022 | 0.5847 | 0.3649 | 0.5787 |
| AdcSR-1 | Partial | 15.37s | 42.91GB | 4.0064 | 3.4083 | 0.6304 | 0.4001 | 0.6523 |
| *GenDR-1 | Pass | 46.51s | 8.47GB* | 3.6740 | 3.8478 | 0.6968 | 0.4199 | 0.6968 |
| **GenDR-Pix** | Pass | 3.58s | 15.10GB | 3.5701 | 3.7172 | 0.6612 | 0.4092 | 0.6570 |

*: inference using tiling strategy for an image.

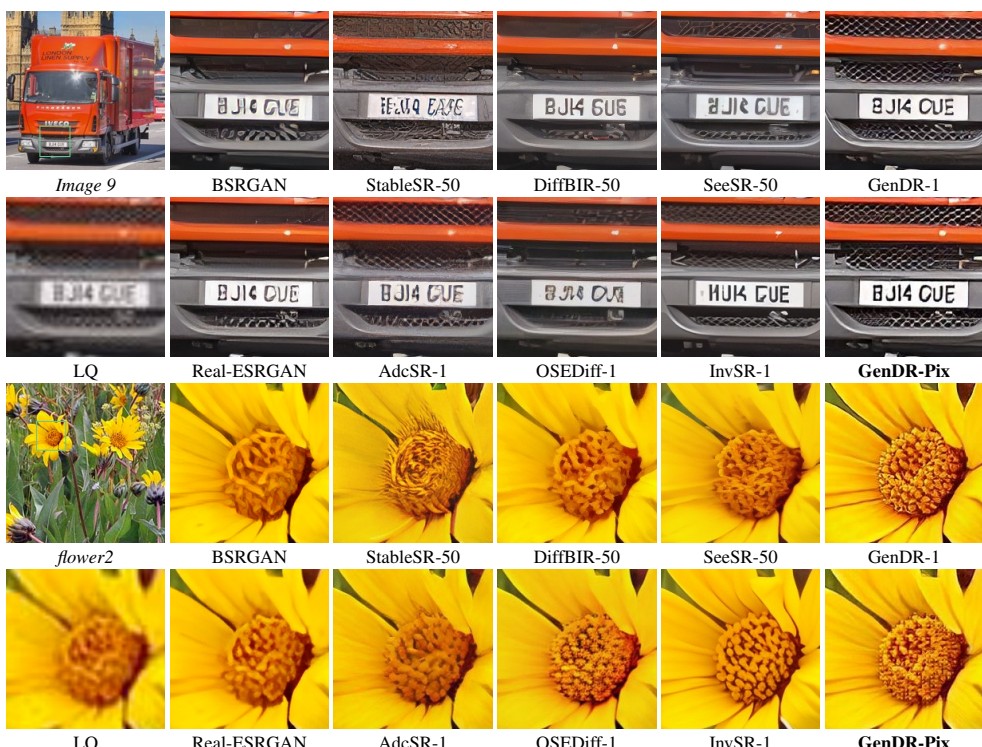

Figure 16: Visual comparison of GenDR-Pix with other methods for ×4 task on RealSet80 dataset.

restores the plate number "BJI4 CUE" and grille. For *flower2*, our method presents more meticulous details for the stamen, despite suffering sequela of repeated patterns.

## A.3 LIMITATIONS

Our method removes the entire VAE of a one-step diffusion-based SR model to chase the best efficiency. In situations of the multiple-step model, while the MFS loss and RandPad still work, the encoder and decoder are inapplicable to be removed through multiple stages.

Another limitation lies in the native unet in SD2.1, which executes most computations under lossy downsampled resolution, ignoring the local high-frequency structure. This may intensify the repeated artifacts. In future work, we will consider structure adaptation based on both UNet and DiT.

This paper is an explorative work to replace VAE with pixel (un)shuffle in the real-world SR task. However, considering our methods being theoretically competent in other low-level tasks, we believe there would be more following work to simplify the diffusion model by removing VAE.

## B    THE USE OF LARGE LANGUAGE MODELS (LLMS)

We use Large Language Models in this research project to assist with various aspects of the writing and research process, including text polishing and refinement, and grammar enhancement.

