# OpenReview forum: "Eliminating VAE for Fast and High-Resolution Generative Detail Restoration"
_ICLR.cc/2026/Conference — ICLR 2026 Poster_

### Official Review · Reviewer_mmPH · 2025-10-31

**Soundness:** 3
**Presentation:** 3
**Contribution:** 3
**Rating:** 6
**Confidence:** 5

**Summary:**

The paper identifies the VAE as the principal latency and memory bottleneck in one-step diffusion SR, then removes it by operating in pixel space with pixel-unshuffle/shuffle. It proposes a two-stage adversarial distillation, a masked Fourier loss to suppress periodic artifacts induced by large-factor shuffling, a random-padding augmentation for discriminator stability, and a padding-based classifier-free guidance that doubles as a lightweight self-ensemble. Experiments indicate about 2.8× speed-up and roughly 60% memory savings versus a GenDR baseline.

**Strengths:**

- The paper provides convincing evidence that VAE dominates both time and memory.

- The artifact analysis in frequency space is insightful, and the masked Fourier loss directly targets the shuffle-periodic spikes observed empirically.

- Efficiency results are strong, including 4K processing without tiling in certain settings, and the ablations for discriminator choice, RandPad and MFS are helpful.

**Weaknesses:**

- My main concern is that the conceptual novelty is incremental relative to prior encoder-removal and pruning approaches although the authors emphasized the differences to AdcSR.
- Its robustness across scale factors and severe degradation is under-explored.
- The effect of RandPad to boundaries is not analysized and discussed.

**Questions:**

See weaknesses.

---

> ### Author Response · Authors · 2025-11-21
>
> We appreciate the reviewer's insightful feedback and careful evaluation of our work. We greatly appreciate the recognition of the strengths in our method, including the efficiency gains and loss design. Below, we try to address the concerns in detail.
>
> 1. **About incremental novelty**. We acknowledge that GenDR-Pix is a deep-in extension based on AdcSR. But we would like to emphasize that the core innovation lies not merely in "removing the decoder" as a standalone step, but also in the systematic pipeline designed to bridge the latent-space and pixel-space to eliminate both the encoder and decoder, which addresses fundamental bottlenecks brought by VAE.
>
> 2. **About results across multiple scale factors**. We agree that the robustness across scale factors and severe degradation should be included. We replenish the $\times$2 and $\times$3  comparison with RealESRGAN, GenDR, and OSEDiff in Table 10 of our revised manuscript. Unlike AdcSR tailored for $\times$4 SR, the proposed method achieves remarkable performance for both $\times$2 and $\times$3 SR, showing its effectiveness and robustness. In terms of serve degradation, we will inlcude more experiment on more challenge task with harder-degradation, e.g., face restoration task and add more visual comparison.
>
> 3. **About effects of RandPad to boundaries**. As shown in Figure 10, we employ reflect mode to apply a random spatial shift for discrinminator input, which bring slightly change to the image boundaries. For the generation model, since it is trained on unpadded images and only the discriminator feedbacks only score, the generator doesn't learn or produce padded boundaries. Moreover, the random shift is wrapped consistently and does not introduce boundary artifacts in practice.

---

### Official Review · Reviewer_occV · 2025-10-31

**Soundness:** 2
**Presentation:** 2
**Contribution:** 3
**Rating:** 4
**Confidence:** 4

**Summary:**

The paper presents GenDR-Pix, a pixel-space adaptation of the one-step diffusion-based super-resolution method GenDR, aimed at addressing the memory and latency limitations of VAE-based pipelines for high-resolution images. By leveraging pixel-(un)shuffle operations, the authors remove the VAE entirely, but note that high upscaling (×8) can introduce repeated-pattern artifacts. To mitigate this, they propose a multi-stage adversarial distillation strategy, using generative features from previous stages to guide discrimination, along with random padding to augment features and prevent discriminator collapse, and a masked Fourier space loss to penalize amplitude outliers. Additionally, a padding-based self-ensemble with classifier-free guidance is integrated to improve inference performance. Experiments show that GenDR-Pix achieves 2.8× faster inference and 60% memory reduction compared to GenDR, with negligible visual degradation, and is capable of restoring 4K images in 1 second using 6 GB of memory, outperforming other one-step diffusion SR methods.

**Strengths:**

1. The idea of achieving efficient high-resolution image restoration by removing the VAE is interesting and promising.

2. The paper is well-written.

**Weaknesses:**

1.Baselines are VAE latent-space-based models, and forcing the removal of the VAE to operate in pixel space introduces certain risks. As shown in Tables 2 and 3, the proposed method does not achieve the best results. The VAE removal strategy requires more thorough analysis and theoretical justification, which is currently lacking in the paper. The authors do not explain the underlying motivation beyond efficiency, nor do they clarify why this approach works in practice.

2. If the goal is to reduce GPU memory consumption, there are simpler and more effective alternatives, such as tiled VAE, which is already commonly used in high-resolution image generation models.

3. The current experiments are conducted only on SD2.1. To fully validate the effectiveness of the proposed method, experiments should be performed on stronger models, such as SDXL or FLUX-based models. A broader range of evaluations is necessary to convincingly demonstrate its effectiveness.

4. If removing the VAE can indeed improve efficiency, why not adopt a non-VAE-based generative model instead?

**Questions:**

No

---

> ### Author Response · Authors · 2025-11-21
>
> We thank the reviewer for raising these important points. Here is our in-depth response:
>
> 1. **About theoretical justification and performance clarification**. Generally, our method is based on the equivalent representation of pixel-space diffusion and latent-space diffusion, which we add related proof in our above response to Reviewer 6W9z. In fact, the representation equivalence simplifies the VAE removal task to be a model distillation/pruning task, which eliminates the VAE, the most costly part in one-step SR diffusion. However, as a distillation/pruning task, removing VAE only involves less than 10% parameters, showing potential to result in a slight performance drop according to previous distillation work.
>
> 2. **About comparison with VAE method**. We try to dispel reviewer's concerns by providing more comprehensive comparisons with both tiled VAE and distilled VAE.
>    - **Comparison with tiled VAE**. Despite tiled VAE can save memory reduction, it would significantly increase the latency. As shown in the following table, GenDR with tiled VAE costs 46s to process a 4K image, which is about 15$\times$ longer than GenDR-Pix (3.6s). Moreover, for GenDR-Pix, we can also apply tiled strategy to further lower the GPU footprint.
>
>       | Method | Latency | Memory | NIQE |  MANIQA | CLIPIQA |
>       | - | - | - | - | - | - |
>       | GenDR-Tiled VAE | 46.51s | 8.47GB | 3.6740 | 0.4199 | 0.6968 |
>       | GenDR-Pix | 3.58s | 15.10GB | 3.5701 | 0.4092 | 0.6570 |
>
>    - **Comparison with distilled VAE**. We compare pixel-(un)shuffle with distilled VAE below. Compared to the distilled VAE heavily damages image quality, the pixel-(un)shuffle is a rearrangement operator that faithfully restores all pixels.
>
>       | VAE-from | Pixel-(un)shuffle | GenDR | GenDR-dstilled | SD 2.1 | TAESD 2.1 |
>       | :- | :-: | :-: | :-: | :-: | :-: |
>       | #Param | 0 | 57.27M | 37.99M | 57.27M | 2.44M |
>       | PSNR (dB) | inf | 30.94 | 28.70 | 27.20 | 26.72 |
>       | LPIPS | 0.0000 | 0.0752 | 0.1035 | 0.1503 | 0.2221 |
>
>        We also compare our removing VAE strategy with distilling VAE methods, where eliminating VAE maintains the perceptual quality as GenDR-distilled VAE and achieves better fidelity than GenDR. Moreover, compared to distilling VAE, removing VAE brings 49% reduction in latency and 45% in memory cost, dominating the effectiveness of removing VAE.
>
>       | Method | Runtime | Memory | PSNR | SSIM | ClipIQA | MUSIQ |
>       | :- | :-: | :-: | :-: | :-: | :-: | :-: |
>       | GenDR | 1.62s | 8.32GB | 23.18 | 0.7135 | 0.7014 | 68.36 |
>       | GenDR-Distilled VAE | 1.03s | 7.57GB | 22.53 | 0.6954 | 0.6851 | 67.37 |
>       | GenDR-Pix | 0.53s | 4.13GB | 25.03 | 0.7513 | 0.6816 | 66.29 |
>
> 3. **About model generality**. Generally, SDXL shares a similar UNet architecture as SD2.1 for noise prediction. Since we do not have an SDXL-based one-step diffusion SR model, we evaluate another UNet-based one-step diffusion model, OSEDiff, for fast validation.
>
>       | Method | VAE | #Runtime | #Memory | MUSIQ |
>       | :- | - | :-: | :-: | :-: |
>       | OSEDiff | Full VAE |1.57s | 24645MB | 69.19 |
>       | OSEDiff-stage 1 | Remove Encoder | 1.08s | 22049MB | 69.07 |
>       | OSEDiff-stage 2 | Remove VAE | 0.32s | 8343MB | - |
>
>
>     Regarding DiT-based methods like FLUX/SD3.5, employing a similar strategy has minimal fittings. For the UNet-based latent diffusion model, the bottleneck of runtime and memory cost is VAE, while for DiT-based LDM, the bottleneck of runtime and memory cost is DiT, leading to the removal of VAE with only marginal benefit. Moreover, the key motivation of GenDR-Pix is to enable the fastest and highest resolution (e.g., 4K) deployment, while DiT is too heavy for practical usage. In the following table, we exhibit the detailed runtime and memory cost for both UNet-Pix based on SD 2.1. and DiT-Pix based on SD3.5.
>
>    | Method | #Runtime | #Memory |
>    | :- | :-: | :-: |
>    | DiT | 571ms | 41141MB |
>    | DiT-Pix | 405ms (-22%) | 40121MB (-3%) |
>    | UNet | 263ms | 4709MB |
>    | UNet-Pix | 73ms (-78%) | 2785MB (-41%) |
>
> 4.  **About non-VAE-based method**. Although more and more pixel-space (non-VAE) diffusion emerges, there is no well-pretrained model for the SR task (most based on ImageNet-256). Regarding GANs, they have been surpassed by one-step diffusion to a large extent. To achieve better performance, it is more conceivable to modify the existing SOTA. We also tried to train from scratch and observed a significant drop in perceptual quality compared to the VAE-based model:
>
>       | Method | NIQE | ClipIQA | MUSIQ |
>       | - | - | - | - |
>       | without guidance | 5.3095 | 57.79 | 0.4685 |
>       | full guidance | 4.8380 | 63.87 | 0.5764 |

---

> > ### Comment · Reviewer_occV · 2025-11-28
> >
> > Thank you very much for the detailed response, which has addressed most of my concerns ( It is also worth noting that I recently contracted influenza A, which caused some delay in my response. ) . I have decided to raise my score to 6. Since OpenReview currently does not allow score updates, I will update it once the system permits, or I will inform the AC directly.

---

> > > ### Author Response · Authors · 2025-11-28
> > >
> > > Thanks for your reply. We are glad that our response dispelled your concern. We sincerely appreciate your feedback, which helps us improve the overall quality of our manuscript.

---

### Official Review · Reviewer_6W9z · 2025-11-01

**Soundness:** 3
**Presentation:** 2
**Contribution:** 2
**Rating:** 6
**Confidence:** 3

**Summary:**

This work aims to eliminate the VAE bottleneck in one-step diffusion-based super-resolution models to enable efficient on-device image restoration. The authors focus on the problem of high latency and memory consumption caused by the VAE in latent-space diffusion models. The authors then try to address this limitation by refining the VAE encoding and decoding, which are often in low efficiency, especially for high-resolution inputs. The authors propose to use a pixel-space diffusion SR model that replaces the VAE encoder and decoder with pixel-level operations and model compression techniques. Evaluations based on some image super-resolution datasets show the effectiveness of the proposed method in terms of restoration quality, inference speedups and memory reduction.

**Strengths:**

Overall, this paper focuses on a meaningful topic in improving the efficiency of the diffusion-based restoration process. The whole idea is easy to understand and seems to be effective. The authors identify the VAE as the primary bottleneck in both latency and memory usage of the diffusion-based SR models. The authors aim to eliminate the VAE to improve efficiency without significantly compromising visual quality. The experimental results show reductions in memory and time costs, making the proposed SR method accessible for real-world tasks.

**Weaknesses:**

I appreciate the authors’ efforts to design a more effective diffusion quantization method. Here, I summarize my major concerns and questions in three parts.

1. The authors propose to replace the VAE with pixel-unshuffle and shuffle operations, so the diffusion procedure is from latent space to traditional pixel space. However, it is unclear about the effectiveness of this strategy. The authors could give a deeper theoretical justification for why latent-space diffusion can be replaced by pixel-space diffusion.
2. The authors try to prevent discriminator collapse and improve generalization by randomly padding images before feeding them into the discriminator. Could you give some ablation studies on how the padding affects the data augmentation performance?
3. In equation 2, the authors utilize the masked Fourier space (MFS) loss based on a band-rejection filter. Could you give more details on the mask initialization and how the mask collaborates with the random padding?

**Questions:**

Please clarify my concerns in the weakness part. I'm not an expert in this field, but I think this paper may have some merits. I would like to check the authors' rebuttal to decide my final rating.

---

> ### Author Response · Authors · 2025-11-21
>
> We thank the reviewer for the constructive feedback and address the three major concerns below.
>
> 1. **About theoretical justification**.  We try to dispel reviewer's concern by replenishing the interpretability why why latent-space diffusion can be replaced by pixel-space diffusion.
>
>    Despite pixel-space diffusion emerging early, latent diffusion models have become the de facto standard due to their effectiveness in reducing computational demands and squeezing generation space, leading to an easier training procedure. However, the representation capabilities of pixel-space and latent-space are equal. Here, we give a simple proof.
>
>     > Assuming the encoder $E$ and decoder $D$ are Lipschitz-continuous and approximately invertible on the data manifold, we can construct a continuous bijection between pixel-space and latent-space noise-prediction functions.
>     >
>     > **Forward mapping**
>     > For any pixel-space predictor $\varepsilon_\theta(x_t,t)$ , define the corresponding latent-space predictor:
>     > $$\varepsilon_\phi(z_t,t) = E\big(\varepsilon_\theta(D(z_t),t)\big). $$
>     > Since $E$, $D$, and $\varepsilon_\theta$ are continuous, this mapping is continuous in $\theta$.
>     >
>     > **Inverse mapping**
>     > For any latent-space predictor $\varepsilon_\phi(z_t,t)$ , define:
>     > $$ \varepsilon_\theta(x_t,t) = D\big(\varepsilon_\phi(E(x_t),t)\big). $$
>     > When $D \circ E \approx \mathrm{Id}$ on the data manifold (a common assumption in diffusion theory), the inverse is well-defined and continuous.
>
>     > These two mappings are continuous and mutually invertible, forming a homeomorphism between the two function classes.
>
>     Thus, pixel-space and latent-space diffusion models are representationally equivalent, implying the two model classes are representationally equivalent. Moreover, recent works, e.g., MDM [a], PixelFlow [b], JIT[c] have validated the effectiveness of pixel-space diffusion in the T2I task. Compared to the T2I task, SR is a more straightforward generation task to solely restore partial missing information, which further reduces training difficulty. These enable us to remove the entire VAE to degrade latent-based GenDR to pixel-space GenDR-Pix.
>
>    *[a] Matryoshka Diffusion Models. ICLR 2024.*
>
>    *[b] PixelFlow: Pixel-Space Generative Models with Flow. ArXiv 2025.*
>
>    *[c] Back to Basics: Let Denoising Generative Models Denoise. ArXiv 2025.*
>
> 2. **About the ablation study on RandPad**. We agree that understanding the effect of random padding is important. Random padding breaks the periodic alignment and forces the discriminator to learn continuous and shift-invariant representations. We conducted targeted ablations in Tab. 4. In summary, adding RandPad improves both stability and perceptual metrics (NIQE: 4.8380 → 4.6648 and ClipIQA: 0.5764 → 0.5937), showing its effectiveness.
>
> 3. **About mask definition of MFS Loss and coherence with RandPad**. Generally, the band-rejection mask \(M\) zeros out a small neighborhood (width \(2s\)) around all frequency coordinates associated with these periodic spikes. Formally:
>
>     $$ M[u,v] = 1, \text{if } u \in [0, H/8, 2H/8, \dots]\pm s \text{ or } v \in [0, W/8, 2W/8, \dots] \pm s;  \text{otherwise}, 0. $$
>
>    Specifically, the $s$ is set as 2 for 512$\times$512 image, to make it penalize only the artifact frequencies while leaving the natural image spectrum untouched.
>    Regarding coherence with RandPad, the mask is used in MFS loss, which directly penalizes the output image to supervise the output image, avoiding generating periodic frequency spikes. It is direct and intuitive to leverage the identity of frequency. On the other hand, RandPad is also based on frequency identity, but we use it to improve the capability and robustness of the discriminator. Overall, MFS directly calibrates the output, and RandPad provides indirect supervision by improving the discriminator.

---

### Official Review · Reviewer_bRU9 · 2025-11-01

**Soundness:** 3
**Presentation:** 3
**Contribution:** 3
**Rating:** 6
**Confidence:** 5

**Summary:**

This paper presents GenDR-Pix, a one-step super-resolution model that removes the VAE encoder and decoder, enabling direct pixel-space generation. To stabilize training and reduce artifacts, the authors introduce a two-stage adversarial distillation framework along with MFS loss, RandPad, and PadCFG. Trained on 20M images, the model achieves faster inference and lower memory usage while maintaining high perceptual quality.

**Strengths:**

- Clear motivation and simplification of architecture:
The paper explores removing the VAE encoder and decoder in one-step super-resolution, which reduces system complexity and avoids reliance on latent-space operations.

- Practical efficiency improvements:
By operating entirely in pixel space, the method achieves notable gains in inference speed and memory efficiency, as supported by quantitative results.

- Targeted technical solutions:
The paper introduces specific techniques (e.g., MFS loss, RandPad, PadCFG) to address known challenges like checkerboard artifacts and unstable guidance in pixel-space generation.

- Empirical validation:
The proposed method is evaluated on multiple benchmarks, with ablation studies that reasonably support the effectiveness of key components.

**Weaknesses:**

none

**Questions:**

Recent one-step RealISR methods such as (e.g., OSEDiff), commonly leverage pre-trained T2I diffusion models  as initialization. With such strong priors, these models require only 10k–20k training steps to adapt to the SR task. In contrast, the authors of this paper move away from this paradigm: they remove both the VAE encoder and decoder, adopt a GAN-based training scheme, and train from scratch on 20M images using adversarial and perceptual losses.

This shift raises a fundamental question: Are pre-trained T2I diffusion models truly necessary for high-quality one-step super-resolution, or can a GAN-based model trained with sufficient data achieve comparable results without them?

More specifically:

- Has the author tried training the student model without any teacher guidance, relying solely on GAN and perceptual losses?
- Has the author evaluated using a randomly initialized teacher, instead of one derived from a pre-trained T2I model?

These experiments would help clarify whether the benefits come primarily from the pre-trained semantic priors of T2I models or from the data scale and the adversarial learning strategy itself.

Another important aspect is the role of training data. The authors employ 20 million images for training, which is significantly larger than the datasets used in previous one-step SR works.

To what extent does the scale and quality of this dataset, as well as the number of training iterations, contribute to the final performance?

For example:

- Has the model been evaluated under reduced data regimes (e.g., 1M or 5M images)?
- How sensitive is the performance to the number of training steps?
- Is the high perceptual quality primarily a result of the large-scale data, rather than architectural or training innovations?

---

> ### Author Response · Authors · 2025-11-21
>
> We appreciate the reviewer's professional and constructive feedback. We would discuss the mentioned question below.
>
> 1. **About the necessity of T2I models as intermediate teacher**. Yes, we believe that directly training a one-step/GAN-based model will be the trend for efficient super-resolution, and it will achieve similar or even better restoration performance. However, as the reviewer mentioned, the data is the core limitation for training from scratch in terms of both quality and quantity. Despite using a 20M image, most of them are filtered from the T2I, whose quality is average and far behind SR datasets like DIV2K or Flickr2K, limiting the performance upper bound of our model to the same level as the T2I-based one. Meanwhile, without the semantic priors provided by the teacher, the model struggles to learn high-frequency details and accurate texture restoration, thereby requiring a longer training period and more data. However, T2I-based models, like OSEDiff, are derived from a well-pretrained model like SD2.1, which has been trained with over 5B data. This enables them to easily achieve better restoration quality with fewer training iterations and partial trainable parameters, like using LoRA. To accelerate training and convergence, we have to use pre-trained T2I diffusion models for high-quality one-step super-resolution and then remove the VAE for acceleration. We conduct a simple ablation study according to the suggested comparison:
>
>    - Without any teacher guidance. We evaluate the influence of teacher through training Stage-II GenDR with varying guidance degrees, including without guidance, adversarial guidance (using the teacher as feature extractor for the discriminator), reference guidance  (using the teacher's results for supervision), and full guidance. In general, the GenDR-pix without guidance achieves similar performance to the reference result-guided model, aligning with our expectation that training without teacher works, but it wouldn't surpass ones with teacher guidance due to limitations of the existing dataset scale and quality.
>
>       | Method | NIQE | ClipIQA | MUSIQ |
>       | - | - | - | - |
>       | without guidance | 5.3095 | 57.79 | 0.4685 |
>       | reference guidance | 4.8343 | 60.95 | 0.4924 |
>       | adversarial guidance | 5.0501 | 62.90 | 0.5505 |
>       | full guidance | 4.8380 | 63.87 | 0.5764 |
>
>    - With a randomly initialized teacher (discriminator). In fact, our discriminator uses a pretrained model for feature extraction and is fixed during training. We train multiple discriminator heads (randomly initialized) to calculate adversarial loss. We compare GenDR-pix with varied extraction model for the second-stage training in Table 4.
>
>
> 2. **About more discussion and ablation study for training data and period**. During the training process, we utilize approximately 20 million LQ-HQ image pairs under a longer training scheduler to optimize GenDR-pix, which surpasses most existing one-step models. In summary, we employ the data and scheduler strategy for two reasons:
>
>     - GenDR-Pix depends on full-parameter training to transfer from latent space to pixel space, which requires more data and time to stabilize the training process and avoid overfitting.
>
>    - GenDR-Pix adopts more complex degradation, which integrates multiple upscale factors using both the RealESRGAN pipeline and the APISR pipeline.
>
>    To directly quantify the influence of dataset and training steps, we visualize performance trajectories in Fig.15 of our revised manuscript. All three curves show non-linear ``phase-transition'' behavior. Specifically, ClipIQA encounters a sharp improvement around 50k–100k iterations. However, more importantly, the absolute performance consistently increases across all regimes, even before these transition points, indicating that the observed gains are not accidental artifacts of data scaling, but a stable consequence of the proposed distillation strategy. Moreover, the performance gap between 1M and 20M images is only 1.15 on MUSIQ, which is significantly smaller than the influence of our distillation strategy.

---

### Author Response · Authors · 2025-12-03

Dear Area Chair,

We deeply appreciate your willingness to dedicate your valuable time to conducting a thorough review of our submission and rebuttal amidst the unforeseen circumstances.

To assist your evaluation, we provide a brief summary of our work.

**Paper Overview**

For one-step diffusion-based SR, VAE is the bottleneck for both runtime and memory, limiting the throughput and high-resolution restoration. Thus, we try to eliminate it by transferring a VAE-based diffusion to a VAE-free model. To ensure the calculation efficiency, we employ the pixel-unshuffle/shuffle operator to replace the encoder/decoder. Similar to the patchifying-based method, the pixel rearrangement will introduce periodic artifacts. To alleviate the distortion, we employ a multi-stage adversarial learning approach with a tailored loss function (MFS loss) and data augmentation (RandPad).


**Initial Ratings**: 6 (bRU9); 6 (6W9z); 4 (occV); 6 (mmPH)

**Consensus on Strengths**:

1. Intuitive motivation and effective strategy (All reviewers).

2. Practical efficiency improvements (bRU9, 6W9z, mmPH).

3. Empirical validation and convincing ablation study (bRU9, mmPH).

**Main Concerns and Rebuttal Actions**:

1. **Extensive discussion for guidance from teacher model (bRU9, occV)** : We conduct experiment to compare (a) without guidance (training non-VAE model from scatch), (b) adversarial guidance (using the teacher as feature extractor for the discriminator), (c) reference guidance (using the teacher's results for supervision), and (d) full guidance ((a)+(b)).

2. **Extensive discussion on dataset and training scale (bRU9)**:  We add a detailed discussion in L845-852 and Fig.15 to show the influence of dataset scale and training interaction.

2. **Theoretical justification (6W9z)**: We include a simple proof for the representation identity between the latent-based and pixel-based models to support that latent-space diffusion can be replaced by pixel-space diffusion.

3. **Comparison with VAE-tile/distillation/pruning. (occV)**: We add a detailed performance comparison with these methods to show our strategy achieves better trade-offs in fidelity and inference cost.

4. **Incremental novelty and more results on multi-scale. (mmPH)**: We clarify the novelty in the systematic pipeline designed to bridge the latent-space and pixel-space. Regarding multi-scale results, we provide a comparison in Table 10 of revised paper.

We sincerely thank all reviewers for their constructive feedback. We believe that our revisions, supplementary experiments, and detailed responses have meticulously addressed all concerns. We kindly request that our rebuttal be fully considered in your final decision, and we are truly grateful for your time and expertise throughout this review process.

Best regards,

Authors

---

### Meta-Review · Area_Chair_3rzg · 2026-01-05

**Summary:**

This work explores a one-step super-resolution diffusion model that removes the encoder and decoder of a VAE while maintaining high-quality visual generation, effectively addressing the efficiency bottleneck in the VAE-based diffusion frameworks. This paper received an initial average rating of 5.5 (4, 6, 6, 6). Reviewers raised concerns regarding the role and necessity of the guidance of teacher model (bRU9, occV), dataset scale and training stability (bRU9), theoretical justification for replacing latent-space diffusion with pixel-space diffusion (6W9z), comparisons with related efficiency-oriented approaches such as VAE-tiling, distillation, and pruning (occV), as well as the incremental novelty of the proposed method and its effectiveness in multi-scale settings (mmPH).

After reading the reviews and the rebuttal, most of these concerns have been adequately addressed, and all reviewers subsequently converged to positive ratings (6, 6, 6, 6). The proposed VAE-free diffusion framework offers significant gains  in efficiency, both in inference speed and memory cost, which is particularly valuable for real-world use, especially on resource-constrained devices. Given the unified positive ratings from the reviewers and the overall contribution, the Area Chair recommends Accept.

**Reviewer Concerns:**

The authors added further experiments, discussions, and results in the rebuttal and revised paper. They clarified the role of teacher guidance, provided additional analyses on dataset scale and training stability, and included a theoretical justification for replacing latent-space diffusion with pixel-space diffusion. The rebuttal also added comparisons with efficiency-oriented baselines and additional results on multi-scale settings, helping to clarify the novelty and practical trade-offs of the proposed approach. Overall, most of the concerns listed above were sufficiently addressed. No major outstanding issues remain.

**Reviewer Scores:**

Reviewers bRU9, 6W9z, and mmPH gave positive initial ratings of 6 and did not raise new major concerns afterward, thus their scores would be expected to remain unchanged.

Reviewer occV indicated during the discussion that his score would be raised from 4 to 6.

Overall, all reviewers converged to positive ratings of 6.

---

### Decision · Program_Chairs · 2026-01-26

Accept (Poster)